# Comparative Study of the Cytokine Profiles of Serum and Tissues from Patients with the Ossification of the Posterior Longitudinal Ligament

**DOI:** 10.3390/biomedicines11072021

**Published:** 2023-07-18

**Authors:** Li-Yu Fay, Chao-Hung Kuo, Hsuan-Kan Chang, Mei-Yin Yeh, Chih-Chang Chang, Chin-Chu Ko, Tsung-Hsi Tu, Yi-Hsuan Kuo, Wang-Yu Hsu, Chien-Hui Hung, Ching-Jung Chen, Jau-Ching Wu, May-Jywan Tsai, Wen-Cheng Huang, Henrich Cheng, Meng-Jen Lee

**Affiliations:** 1Institute of Pharmacology, National Yang Ming Chiao Tung University, No. 155, Sec. 2, Linong St., Taipei 11217, Taiwan; leofay1978@gmail.com (L.-Y.F.); hansamu0627@gmail.com (C.-C.K.); jauching@gmail.com (J.-C.W.); hc_cheng@vghtpe.gov.tw (H.C.); 2School of Medicine, National Yang Ming Chiao Tung University, No. 155, Sec. 2, Linong St., Taipei 11217, Taiwan; chaohungk@gmail.com (C.-H.K.); hsuankanchang@gmail.com (H.-K.C.); ccchang74@gmail.com (C.-C.C.); thtu0001@gmail.com (T.-H.T.); b101094018@tmu.edu.tw (Y.-H.K.);; 3Department of Neurosurgery, Neurological Institute, Taipei Veterans General Hospital, No. 201, Sec. 2, Shipai Rd., Taipei 11217, Taiwanmjtsai2@vghtpe.gov.tw (M.-J.T.); 4Institute of Biomedical Sciences, Academia Sinica, and National Yang Ming Chiao Tung University, No. 155, Sec. 2, Linong St., Taipei 11217, Taiwan; 5Neural Regeneration Laboratory, Department of Neurosurgery, Neurological Institute, Taipei Veterans General Hospital, No. 201, Sec. 2, Shipai Rd., Taipei 11217, Taiwan; 6Department of Applied Chemistry, Chaoyang University of Technology, 168, Jifeng E. Rd., Taichung 413310, Taiwan

**Keywords:** cervical spondylotic myelopathy, spinal cord injury, ossification of the posterior longitudinal ligament, osteoprotegerin, angiogenin, osteopontin, leptin, cytokine array

## Abstract

Background: The ossification of the posterior longitudinal ligament (OPLL) is one of the contributing factors leading to severe cervical spondylotic myelopathy (CSM). The mechanism causing ossification is still unclear. The current study was designed to analyze the specimens of patients with or without OPLL. Methods: The study collected 51 patients with cervical spondylosis. There were six serum samples in both the non-OPLL (NOPLL) and OPLL groups. For tissue analysis, there were seven samples in the NOPLL group and five samples in the OPLL group. The specimens of serum and tissue were analyzed by using Human Cytokine Antibody Arrays to differentiate biomarkers between the OPLL and NOPLL groups, as well as between serum and OPLL tissue. Immunohistochemical staining of the ligament tissue was undertaken for both groups. Results: For OPLL vs. NOPLL, the serum leptin levels are higher in the OPLL group, corroborating others’ observations that it may serve as a disease marker. In the tissue, angiogenin (ANG), osteopontin (OPN), and osteopro-tegerin (OPG) are higher than they are in the OPLL group (*p* < 0.05). For serum vs. OPLL tissue, many chemotactic cytokines demonstrated elevated levels of MIP1 delta, MCP-1, and RANTES in the serum, while many cytokines promoting or regulating bone genesis were up-regulated in tissue (oncostatin M, FGF-9, LIF, osteopontin, osteoprotegerin, TGF-beta2), as well as the factor that inhibits osteoclastogenesis (IL-10), with very few cytokines responsible for osteoclastogenesis. Molecules promoting angiogenesis, including angiotensin, vEGF, and osteoprotegerin, are abundant in the OPLL tissue, which paves the way for robust bone growth.

## 1. Introduction

Cervical spondylotic myelopathy (CSM) is a degenerative disease caused by hypertrophy of the ligaments and bony structures due to biomechanical stress. The ossification of the posterior longitudinal ligament (OPLL) is one of the most severe causes of CSM. OPLL leads to the direct compression of the spinal cord and may induce other reactions. The multifactorial pathogenesis of OPLL and CSM merges at the location [1,2,3]. The clinical symptoms and signs include sensory deficits, paresthesia, ataxia, and unsteady gait. Spinal cord injury (SCI) is one of the most devastating conditions of CSM, which results in severe motor and sensory dysfunction below the level of injury. It causes primary injury to both neuronal axons and myelin sheaths, which is followed by secondary injury and reactive gliosis, which leads to further neurological damage. The secondary injury after SCI involves the release of cytotoxic factors, tissue edema, decreased blood flow, and accelerated apoptosis. Because of the limited regenerative capability and the presence of inhibitory materials in the injury site, effective treatments for SCI are not currently available.

OPLL will lead to heterotopic bone formation at PLL. The process of OPLL includes hypervascular fibrosis, focal calcification, cartilaginous cell proliferation, and, finally, ossification [1,2,3,4]. The pathogenesis of OPLL is an idiopathic and a multifactorial process, which is not fully elucidated. Osteoprotegerin (OPG) is thought to be highly associated with the regulation of bone mass by the inhibition of osteoclasts [5,6]. The pathway may be regulated through Wnt/β-catenin or canonical cascades for bone biology [7,8]. Niu et al. conducted a study in 2017 to compare serum biomarkers in patients with diseases with excessive ossification structures. Serum osteocalcin (OSC) and Dickkopf-related protein 1 (DKK-1) were found to be higher in patients with rheumatoid arthritis, OPLL, and ossification of the ligamentum flavum [9]. In the literature, there was no study regarding tissue biomarkers. The current study was the first study designed to examine possible serum and tissue biomarkers of the ossification process.

## 2. Methods

### 2.1. Patients

A total of 51 samples were collected from 18 cervical OPLL patients who underwent anterior cervical decompression and fixation surgery, along with 33 Non-OPLL (NOPLL) controls; the majority enrolled into this study had cervical degenerative disorders. The diagnosis of OPLL at the cervical spine was confirmed by computer-assisted tomography (CAT) scans, plain radiography, and magnetic resonance imaging (MRI) of the cervical spines (Figure 1). We collected samples of patients who received surgery from April 2018 to May 2019. Patients with neurological deficits, weakness, paresthesia, or gait disturbance would be included. Patients who had previous surgeries, cerebrovascular accidents, coronary artery diseases, autoimmune diseases, or other major systemic diseases would be excluded. We decided whether the patients had OPLL or NOPLL mainly by CAT scans (Figure 1).

The standard right anterior approach for anterior cervical discectomy or corpectomy was conducted in 51 patients. The level and length of incision were confirmed by intra-operative C-arm fluoroscopy. Discectomy would be carried out first. Bilateral uncovertebral joints decompression was performed in the following step. The bone spurs were removed extensively with high-speed burrs or a Kerrison bone punch to achieve generous decompression of the dural sac and exiting nerve roots. The posterior longitudinal ligament (PLL) was resected by a Kerrison punch to ensure thorough decompression. At this step, we would keep the tissue sample of PLL for lab analysis. After meticulous endplate preparation, the implant of a proper size would be inserted to stabilize the cervical spine. In addition, we used copious saline irrigation to clean the operation site before wound closure. We built a database of our institute to prospectively collect our patients’ clinical data at planned pre- and post-operative timepoints, which included radiological and clinical evaluations. The clinical parameters for these patients were evaluated by a questionnaire for Japanese Orthopedic Association (JOA) scores. The protocol was performed by special nurses under the supervision of physicians. This study was approved by the institutional ethics committee of Taipei Veterans General Hospital. Informed consent was obtained from the patients and family members prior to their enrollment in the study.

### 2.2. Spinal Ligament and Serum Samples

During the anterior approach of the cervical decompression surgery, posterior longitudinal ligament specimens were aseptically and carefully harvested and rinsed with phosphate-buffered saline. All patients in the OPLL group underwent anterior cervical corpectomy to decompress the spinal cord, and the OPLL specimens were retrieved after drilling and by microsurgical dissection to ensure both maximal patient safety and tissue harvest. For comparison, non-ossified ligamentous tissue was harvested from each patient of the NOPLL group via anterior cervical discectomy, and then the surrounding tissues, including discs and osteophytes, were carefully dissected away. Blood was collected during the operation and later centrifuged to yield serum.

### 2.3. Histologic Examination

All the NOPLL and OPLL specimens were fixed in 10% neutral-buffered formalin. The samples were axially cut and embedded in paraffin for sectioning. Thin serial sections in 4 µm were obtained, and hematoxylin and eosin (HE) staining was performed to characterize the overall OPLL features. The sections were also subjected to Masson’s Trichrome (MT) staining for the analysis of elastic fibers and immunohistochemistry to detect osteoprotegerin. 

### 2.4. Cytokine Arrays

We analyzed a panel of protein expressions of serum or tissue from NOPLL and OPLL patients using a membrane-based Human Cytokine Antibody Array kit (catalog number ab133998, Abcam, Cambridge, UK), according to the manufacturer’s instructions. The PLL specimens were chopped into 100 mg fragments and frozen at −80 °C. They were ground with mortar and pestles immediately after removal from −80 °C, and 1 mL lysis buffer containing 40 mM Tris buffer (pH 7.5), 8 M urea, 4% CHAPS, 1 mM PMSF, 1 mM Na3VO4, 1 mM dithiothreitol, and a protease inhibitor kit (BM, Germany) was added to extract proteins. For serum samples, neat serum was directly applied to the membrane, undiluted. The protein was quantified by a Biorad protein assay reagent, based on the method of Bradford (1976). Then, 200 μg proteins of the PLL lysates or 1 mL neat serum were analyzed. The relative expression concentrations of 80 soluble human proteins were determined. The proteins are known to be associated with bioactivities such as inflammation, angiogenesis, growth, and ossification processes. This list includes ENA-78, GCSF, GM-CSF, GRO, GRO-alpha, I-309, IL-1alpha, IL-1beta, IL-2, IL-3, IL-4, IL-5, IL-6, IL-7, IL-8, IL-10, IL-12 p40/p70, IL-13, IL-15, IFN-gamma, MCP-1, MCP-2, MCP-3, MCSF, MDC, MIG, MIP-1beta, MIP-1delta, RANTES, SCF, SDF-1, TARC, TGF-beta1, TNF-alpha, TNF-beta, EGF, IGF-I, Angiogenin, Oncostatin M, Thrombopoietin, VEGF-A, PDGF-BB, Leptin, BDNF, BLC, Ckß8-1, Eotaxin, Eotaxin-2, Eotaxin-3, FGF-4, FGF-6, FGF-7, FGF-9, Flt-3 Ligand, Fractalkine, GCP-2, GDNF, HGF, IGFBP-1, IGFBP-2, IGFBP-3, IGFBP-4, IL-16, IP-10, LIF, LIGHT, MCP-4, MIF, MIP-3 alpha, NAP-2, NT-3, NT-4, Osteopontin, Osteoprotegerin, PARC, PLGF, TGF-beta2, TGF-beta3, TIMP-1, and TIMP-2. There were six serum samples in both the NOPLL and OPLL groups. For tissue analysis, there were seven samples in the NOPLL group and five samples in the OPLL group. The black dots that were revealed were analyzed using an ImageJ protein array analyzer. The intensity of each dot was normalized to that of a positive control dot on each membrane.

### 2.5. Comparison of Serum and Tissue Levels 

The protein expression level of the cytokine array was analyzed by ImageJ software (NIH). The individual levels of the proteins were normalized against the level of the positive control of the cytokine array. The normalized levels of the same proteins for the serum and for the tissue were further analyzed. The ratio of the level of the serum over that of the tissue was obtained, and any value that was more than two was selected (Section 3.6 Levels of Proteins in Serum vs. Tissue: Serum/Tissue > 2). The ratio of the level of the tissue over that of the serum was obtained, and any value that was more than two was selected (Section 3.7 Levels of Proteins in Serum vs. Tissue: Tissue/Serum > 2).

### 2.6. Western Blotting

The total protein was obtained from the PLL specimens, as described for the cytokine array above. Then, 5 μg proteins of the cell lysates were analyzed by a Western blot, using 8% or 12% sodium dodecyl sulfate-polyacrylamide gel electrophoresis (SDS-PAGE) gels. After electrophoresis, the gels were transferred to PVDF membranes (Millipore Corp., Burlington, MA, USA) and incubated overnight at 4 °C with antibodies against the osteoprotegerin antibody (Abcam, Cambridge, UK) and runt-related homeobox (RUNX) (GeneTex, Irvine, CA, USA), followed by a horseradish peroxidase-conjugated secondary antibody for 1 h at room temperature. Immunoreactivity was visualized by enhanced chemiluminescent detection (Perkin Elmer Co., Waltham, MA, USA).

### 2.7. Immunohistochemistry

The sections were deparaffinized and treated with Antigen Unmasking Solution (Vector, Burlingame, CA, USA) for 30 min at 100 °C. Subsequently, these sections were washed with phosphate buffered saline (PBS), and endogenous peroxidase activity was blocked by incubation in methanol containing 0.3% H_2_O_2_ for 30 min. After blocking with 2% horse serum (Vector) for half an hour, the sections were incubated overnight with primary antibodies against osteoprotegerin (Abcam, Cambridge, UK), alkaline phosphatase (GeneTex), angiogenin (Abcam), and osteopontin (Abcam). The sections were then incubated with biotinylated anti-rabbit IgG antibody (Vector) for 60 min. The ABC complex system (Vector) was used to subsequently develop peroxidase via incubation in a VIP solution (Vector) to visualize the peroxidase label. Images were obtained with a microscope equipped with a Charge-coupled Device (CCD) camera (AxioCam HRC, Zeiss, Jena, Germany) to detect the signal. 

### 2.8. Statistical Analysis

Data were analyzed using the mean ± standard error of the mean (SEM). A Student’s *t*-test and Wilcoxon test were used to compare the data between the OPLL and NOPLL groups. For parametric scores, we used the Student-t for comparing patients’ ages and protein concentrations. A Wilcoxon test was applied to compare non-parametric scores such as clinical outcomes. The cytokine levels in the serum sample were tested by Šídák’s multiple comparisons test. A *p* value < 0.05 was considered statistically significant.

## 3. Results

### 3.1. Patient Demographics

Specimens of 51 patients were collected; 33 of them were with NOPLL and 18 were with OPLL. The mean age of the NOPLL patients was 54.7 ± 10.2, and that of the OPLL patients was 59.7 ± 9.7 (Table 1). The mean age between groups was similar (*p* = 0.09). The male-to-female ratio was also similar in the two groups (*p* = 0.08). There were six serum samples in both the NOPLL and OPLL groups. For tissue analysis, there were seven samples in the NOPLL group and five samples in the OPLL group. The demographics, including the levels of involvement, the K-line classification of the spine (disease severity), and the types of OPLL, and the clinical evaluations of both the NOPLL and OPLL groups of patients are presented (Table 1). We opted to record the K-line classification and types of OPLL to depict the severity of the disease. In contrast, most of the NOPLL patients had simple, frequently one-level, cervical disc herniation, while the OPLL patients had disease involvement of two or more levels. Among the 18 OPLL patients, 13 had a positive K-line before the surgery, and 10 had mixed-type OPLL. Moreover, all patients of the OPLL group had confirmation of the diagnosis of OPLL by pre-operative CT scans and intra-operative surgical findings (calcified PLL). In contrast, the NOPLL patients had little ossification of the PLL. 

### 3.2. Histological Staining 

All patients in the OPLL group underwent anterior cervical corpectomy to decompress the spinal cord, and the OPLL specimens were harvested. Non-ossified ligamentous tissue was harvested from each patient of the NOPLL group via anterior cervical discectomy. Apart from diagnosis using the medical images, the tissue types were further verified with HE and MT staining (Figure 2). The intact structure of elastic fibers in NOPLL tissue was observed (Figure 2A). The elastic fibers were loose or damaged in OPLL tissue (Figure 2B). Some neovascularization was suspected (Figure 2B).

### 3.3. Serum Cytokine

The serum of both groups was collected and analyzed by Human Cytokine Antibody Arrays. Most of the serum cytokines were similar between the OPLL and NOPLL patients. The 80 cytokines were associated with inflammation, angiogenesis, the growth factor, and ossification (Figure 3). Among them, only the leptin protein levels were significantly different between the OPLL and NOPLL groups. The mean value of leptin was 0.40 ± 0.25 in the NOPLL patients and 1.23 ± 0.23 in the OPLL patients (Table 2, *p* = 0.0138). The level of serum leptin was significantly higher in the OPLL patients.

### 3.4. Tissue Cytokine 

Human Cytokine Antibody Arrays analyzed the biomarkers in the tissues of both groups (Figure 4). The ANG, OPN, and OPG were found to be significantly higher in the OPLL group than in the NOPLL group (3.0 vs. 1.94, 1.81 vs. 1.27, and 1.35 vs. 0.75, *p* < 0.05, respectively, Table 2). 

### 3.5. Level of Proteins in OPLL vs. NOPLL 

Human Cytokine Antibody Arrays analyzed the biomarkers in the tissues of both groups (Figure 3). The ANG, OPN, and OPG were found to be significantly higher in the OPLL group (3.0 vs. 1.94, 1.81 vs. 1.27, and 1.35 vs. 0.75, *p <* 0.01 and *p* < 0.001, respectively, Table 2).

### 3.6. Levels of Proteins in Serum vs. Tissue: Serum/Tissue > 2 

The protein expression level of the cytokine array for serum and for tissue was analyzed and normalized against their own positive control. The ratio of serum/tissue that was more than two was selected and is shown in Table 3. The ratio of tissue/serum that was more than two is shown in Table 3

### 3.7. Levels of Proteins in Serum vs. Tissue: Tissue/Serum > 2 

The protein expression level of the cytokine array for serum and for tissue was analyzed and normalized against their own positive control. The ratio of tissue/serum that was more than two is shown in Table 4.

### 3.8. Verification of Key Protein Expression with Western Blot Methods 

The protein expression level of osteoprotegerin (OPG) was verified using Western blots. Runx2, a key transcription factor that was important during bone generation, was examined for its levels. The level of OPG is significantly higher in the tissue of OPLL patients than that in the non-OPLL patients (Figure 5). 

### 3.9. Immunohistochemical Stain

The IHC staining demonstrated that ossification markers such as ALP, OPG, ANG, and OPN were indeed present in the OPLL tissue and less present in the non-OPLL tissue (Figure 6B,D,F,G vs. A,C,E,G). OPG immunoreactivity was restricted to blood vessel-like structures (Figure 6D), while ANG and OPN were more diffuse, although they appeared to be confined to the general area around the blood vessels-like structure (Figure 6F,H). 

## 4. Discussion

### 4.1. Pathology and Etiology

OPLL is a disease of heterotopic ossification of the posterior longitudinal ligament of the spine. Ectopic ossification can cause severe compression of the neural structure. In the cervical spine, the compression would cause severe neurological deficits [2,3,10]. The pathogenesis of OPLL is a complex interaction comprising genetic and environmental factors [11]. The overproduction of collagen VI and XI would lead to extracellular scaffold formation. Osteoblasts and chondrocytes would proceed to ectopic bone formation or ligament ossification. Bone morphogenic proteins type-2 (BMP-2) could induce osteogenic differentiation in cultured fibroblasts. A histochemical study of surgical specimens from patients with OPLL demonstrated that BMP-2 was present in the ossified matrix, chondrocytes, and fibroblasts in the cartilaginous area near OPLL tissue [12]. 

### 4.2. Severity of the Disease and Importance of the Study

Patients with OPLL were likely to have a comorbid disability. In Taiwan, an 11-year nationwide database study demonstrated that the overall incidence of hospital admission for cervical OPLL-related symptoms was 6.1 per one million person-years [10]. Among the 1651 patients admitted, 85 of them (5.2%) had severe disabilities, and 1109 of them (67.2%) had to have either single- or multi-stage spine surgery. Another study was conducted to evaluate the risks of SCI-related disabilities in patients with OPLL. Data on a total of 5604 patients were collected to compare patients with or without OPLL who received conservative treatment. The incidence rate of cervical SCI in patients with OPLL was significantly higher than that in patients without OPLL. The hazard ratio was as high as 25.64 [3]. Therefore, cervical SCI and related disabilities are more likely to occur in OPLL patients and should be cautioned for special care. 

### 4.3. Leptin

Dr. Ikeda et al. collected data on 125 patients (68 males and 57 females) for serum protein analysis. It was found that hyperleptinemia, hyperinsulinemia, and the female sex are associated with the development of OPLL [13], Another study, in 2005, demonstrated the possibility of polymorphisms of the nucleotide pyrophosphatase gene and the leptin receptor gene, predisposing to an increased frequency and severity of OPLL [14]. Our data demonstrated an association between serum leptin and OPLL. The serum leptin level was significantly higher in patients with OPLL (*p* = 0.0138) than in the non-OPLL patients. 

With our method of analysis, the ratio of serum to tissue for leptin is 8.79 (Table 3), which demonstrated that, although a high leptin level is detected in the serum, this did not translate into the levels in PLL tissues. The levels of leptins are not different between OPLL tissue and non-OPLL tissue (Figure 4). It is therefore possible that, although a higher leptin level is associated with the OPLL incidence, the mechanism of bone generation was not so much regulated by the quantity of leptin per se, locally. Anyhow, the serum leptin difference remains relevant. It is possible that the ossification could be regulated by the leptin receptor activity level in the tissue or the signaling downstream to the receptor. As the leptin level is in proportion to the adipose tissue, it is also possible that the high and differential serum levels are secondary to the excess amount of the adipocyte that was present in people with metabolic syndrome X, which contributes to ossification via other mechanisms such as pressure to the spinal cord.

### 4.4. Angiogenesis Factors: ANG, vEGF, and Osteopretegerin

Bone growth or regeneration depends on vascularization to deliver nutrients, growth factors, minerals, and oxygen for the tissue [15,16]. Angiogenic factors such as the angiogenin and vascular endothelial growth factor (VEGF) have been known to increase during new bone formation. Samples from patients who underwent distraction osteogenesis and spine osteotomy both demonstrated an elevation in serum ANG and other enzymes [17]. In our study, ANG was found to be significantly higher in samples from OPLL patients than in samples from NOPLL patients (*p* = 0.0012) (Table 2). OPG is important for angiogenesis in that it could be induced by integrin and is a survival factor for endothelial cells [18]. OPG combined with FGF-2 promoted vessel formation in vivo [19]. Our data demonstrated that the OPG levels in OPLL were much higher than the serum levels (Table 4), and the OPLL OPG level is significantly higher than that of NOPLL (Table 2). This indicated that the mechanism of bone formation in the OPLL is involved in the upregulation of OPN, which may involve angiogenesis. 

### 4.5. Systemic Inflammation cf Local Inflammation 

The expression levels of several proteins that were related to inflammatory reactions were very different between the serum and the OPLL tissue. This includes MCP-1, RANTES, MIP-1 delta, PDGF-BB, BDNF (higher expression in serum) (Table 3), and IL-10 (higher expression in tissue) (Table 4). 

The chemokine monocyte chemoattractant protein 1 (MCP1) is known as C-C motif ligand 2 (CCL2). It is associated with recruitments of monocytes and T cells to the sites of inflammation [20,21]. MCP-1 and RANTES act as an autocrine loop in human osteoclast differentiation [22]. High levels of blood glucose affect methylation within the MCP-1 promoter region, upregulate MCP-1 levels in the blood, and cause vascular complications in type 2 diabetes [23].

RANTES is also known as chemokine (C-C motif) ligand 5 (CCL5). It is a proinflammatory chemokine. It is mainly expressed by T-cells and monocytes, epithelial cells, and fibroblasts, and it is chemotactic for T cells, monocytes, natural-killer (NK) cells, dendritic cells, and mast cells [24,25]. RANTES is expressed ‘late’ (3–5 days) after T-cell activation [25] and is involved in maintaining inflammation, instead of initiation.

The Macrophage Inflammatory Proteins (MIP) delta belongs to the family of chemotactic cytokines known as chemokines. Members of this family are produced by macrophages and monocytes after they are stimulated with bacterial or proinflammatory cytokines such as IL-1β [26].

Both MCP-1 and RANTES regulate multiple aspects of physiology and pathology; therefore, it is difficult to assign one single aspect to this relatively high level in serum. However, it is fair to say that a certain level of inflammation reaction is detected for these OPLL patients. Interestingly, the classical ‘pro-inflammatory cytokines’ such as Il-1 and TNF-alpha are not highly expressed in serum, as our data demonstrated. Since the role of RANTES is regarded as ‘maintaining’ inflammation, we may say that a chronic inflammatory state is observed in the serum of our OPLL patients. The role of MIP-1 delta is interesting. Apart from being pro-inflammatory, its role of promoting MSC migration is worth noting, as the recombinant chemokines MIP-1δ and MIP-3α induced MSC migration, and anti-MIP-1δ and anti-MIP-3α antibodies added to Huh-7 CM decreased MSC migration [27]. Furthermore, as the proinflammatory cytokines such as IL-1 and TNF-alpha are not highly expressed in the tissue, this suggested that the tissue was not undergoing a full-blown inflammatory reaction such as that of arthritis. MCP-1, RANTES, and MIP1 delta happen to be more ‘chemotactic’ cytokines, and whether they promote stem cell migration into the ligament to form bones is intriguing.

Kawaguch et al. [28] stated that the mean high-sensitivity CRP (hs-CRP) (hs-CRP) in the OPLL group was higher than that in the controls. The mean hs-CRP in the progression group was higher than that in the non-progression group. The hs-CRP test measures the general levels of inflammation in the body, and the patients have different levels up to 2 years. On the other hand, we collected our patient samples one day before the surgery, and we detected differential expression for cytokines typically elevated during the wound or inflammation. This suggested that the body’s general environment undergoes chronic inflammation in OPLL patients. Our data corroborate previous findings and add more information to the inflammatory state of patients

### 4.6. Platelet-Derived EGF, PDGF-BB, and BDNF

EGF, PDGF-BB, and BDNF were regarded as regenerating and growth-promoting cytokines. The relatively higher levels of these proteins in serum may not simply be a regenerative condition. Rather, this author proposes that it is possible that, although angiogenic markers were highly detected in the OPLL tissue, the blood vessel was not of a mature morphology and hence lacks the blood component. The platelets provide rich sources of the aforementioned factors, and the lack of blood or platelets may result in the relatively lower yield of EGF, PDGF-BB, and BDNF in the tissue. 

### 4.7. Bone Formation-Promoting Proteins

Many cytokines that promote osteogenesis or regulate bone metabolism are expressed in higher levels in the OPLL tissue. This includes oncostatin M, FGF-9, LIF, osteopontin, osteoprotegerin, TGF-beta2, and IL-10 (inhibits osteoclastogenesis) (Table 4). 

Oncostatin M, or OSM, is a member of the interleukin 6 family of cytokines [29]. Apart from its role in liver development, hematopoiesis, inflammation, and CNS development, it is also associated with bone formation and destruction [30,31].

FGF family members possess broad mitogenic and cell survival activities. FGF-9 stimulates chondrocyte proliferation [32]. This function may be related to angiogenesis, as FGF9 heterozygous mutant mice had a compromised bone repair after an injury, with lesser expression of VEGF and VEGFR2 and lower osteoclast recruitment [32]. 

Leukemia inhibitory factor, or LIF, is an interleukin 6 class cytokine that affects cell growth by inhibiting differentiation. LIF is typically added to a stem cell culture medium to reduce spontaneous differentiation. The removal of LIF pushes stem cells toward differentiation [33,34]. It promotes growth and differentiation, and the target cells include an influence on bone metabolism and inflammation [35,36].

Interleukin 10 (Il-10) is an anti-inflammatory cytokine in general. IL-10 potently inhibited the RANKL-induced expression of the master regulator of osteoclastogenesis NFATcl and suppressed RANKL-induced gene expression and osteoclast differentiation [37]. 

The ligands of the TGF-beta family bind various receptors and recruit and activate SMAD family transcription factors that regulate gene expression. The TGF-β signal in osteoblasts, skeletal development, bone formation, homeostasis, and disease [38]. It has fundamental roles in both embryonic skeletal development and postnatal bone homeostasis.

Macrophage colony-stimulating factor (M-CSF), also known as the colony stimulating factor 1 (CSF1), is released by osteoblasts as a result of parathyroid hormone stimulation and exerts paracrine effects on osteoclasts [39]. It is involved in the proliferation, differentiation, and survival of monocytes, macrophages, and bone marrow progenitor cells, which are sources of precursors of osteoclasts. RANKL and M-CSF could induce several intermediate stages of differentiation, from hematopoietic stem cells/precursors to tissue macrophages, to differentiate to osteoclasts [40].

It is unknown whether the ligament needs to undergo chondrogenesis before it has full-blown bone formation or what causes further ectopic bone growth, which is the most undesirable result. During this process, there may be several proteins that are critical to the transformation, and these could be the drug targets for preventing or even treating OPLL. Our paper demonstrated that many proteins expressed in high concentrations in the OPLL tissue are either cytokines that promote bone formation (oncostatin M, FGF-9, LIF, TGF-β2), osteoclastogenesis blockers (IL-10), or bone growth markers (OPN, OPG). This demonstrated that the tissue collected was undergoing full-blown bone growth and were at the end stage of bone conversion from the stem cells. The inclination to form bone is so strong that we observed so many cytokines that promote bone formation, and only one (M-CSF) that is possibly promoting osteoclastogenesis. The balance of osteoblasts and osteoclasts is important in maintaining the shape and strength of a healthy bone. This demonstrated that not only was the ligament transformed into bone-rich tissue, but the growth of these bone structures was rather robust, and the balance was tilted toward bone growth and much less bone resorption.

### 4.8. Osteopontin (OPN)

Our data demonstrated that the tissue OPN is more in the OPLL than in the NOPLL patients, which indicated that the mechanism of bone formation in the OPLL indeed involved the upregulation of OPN, which is a component of the bone. OPN is an extracellular structural protein and is a component of the bone. In bone, OPN is expressed mostly by osteoblasts, osteocyctes, and osteoclasts [41]. Runx2 (Cbfa1) transcription factors are required for the expression of OPN [42]. OPN expression could be induced by exposure to pro-inflammatory cytokines, hyperglycemia, and hypoxia [43,44]. It regulates the normal mineralization of bones and teeth [45], as well as ectopic vascular calcification [46,47]. Patients with idiopathic hip OA were reported to have an increase in plasma OPN [48], which is different from our data indicating that the level of serum osteopontin was similar between the OPLL and non-OPLL patients (Figure 3). This suggested that the bone degeneration was not of a significant extent in the OPLL patients and that the osteopontin content of the bone was not leaked into the blood. 

### 4.9. Osteoprotegerin OPG 

OPG is produced by osteoblasts and is a decoy receptor for the receptor activator of the nuclear factor kappa-B ligand (RANKL). When OPG binds to the RANKL, the RANKL/RANK pathway is blocked to inhibit the osteoclastgenesis [5,6]. RANK–RANKL binding activates the nuclear factor kappa B (NF-κB) pathway, resulting in the upregulation of the nuclear factor of activated T-cells, cytoplasmic 1 (NFATc1), and downstream pro-inflammatory cytokines such as TNF-alpha [49,50]. OPG was significantly higher in tissues from OPLL patients than in tissues from NOPLL patients (*p* = 0.0006) (Figure 4 and Table 2). It is six times higher than that detected in the serum (Table 4). The serum OPG levels were not significantly different in NOPLL and OPLL patients, which was consistent with the results of the study conducted by Dr. Niu et al. in 2017 [9]. However, in the OPLL, inflammatory cytokines such as IL-1β, IFN-γ, TNF-α, and TNF-β in the tissue were not significantly different in NOPLL and OPLL patients in the current study (Figure 2 and Figure 3), nor were they higher than the level in the serum. This corroborated our observation that there is no full-blown inflammatory reaction in the serum or in the OPLL (see above; Section 4.5 Systemic Inflammation of Local Inflammation). The high local OPN level regulates an inhibition of osteoclastogenesis and favors the osteoblstogenesis over osteoclastogenesis for bone formation in the OPLL. 

However, although OPN and OPG are both important for the formation of the bone in OPLL tissue, they were also important molecules regulating normal bone metabolism; therefore, they may not be favorable choices for drug targets to act on, as restricted application only to the susceptible ligament may be very difficult. 

### 4.10. IGF-1 and IGFBPs

Growth hormone (GH) and its mediator, the insulin-like growth factor-1 (IGF-1), regulate somatic growth. IGF-1 is a critical mediator of bone growth. It is part of the hypothalamic/pituitary somatotrophic axis that peaks during puberty, stimulates extracellular matrix production, and increases bone density [51,52]. 

IGFBP2 is known to bind IGF and regulate its bioavailability [53]. IGFBP2 may function during normal and diseased metabolism [54]. IGF Binding Protein 2 (IGFBP2) is regulated by leptin with a similarly high potency. The overexpression of IGFBP2 by an adenovirus reversed diabetes in insulin-resistant ob/ob and diet-induced obese mice [55]. In obese patients or patients with type II diabetes, circulating IGFBP-2 levels are low, while the overexpression of IGFBP-2 protects against the disease by inhibiting adipogenesis and modulating insulin sensitivity [54]. 

Although the IGF-1 level was not high in the OPLL tissue, the local IGFBPs are significantly reduced in the OPLL tissue (Table 3) when compared to the serum level. This created an environment or a niche where the IGF-1 could be more available. Increasing or supplementing the local tissue IGFBP2 may prove to be a feasible drug target, as leakage to the blood circulation would probably not affect the systemic concentration. 

### 4.11. IL-6

An increase in IL-6 in the cultured cells derived from the OPLL was observed, and when IL-6 was added to the culture, the levels of proteins such as Sox9, Runx2, and SIRT1 were increased [56]. These proteins are essential for chondrogenic or osteoblast development [56,57]. IL-6, along with IL-1a, FGF-2, and RANTES, was also upregulated in cultured cells from the Ligamentum Flavum of OPLL patients compared to those in CSM patients [58]. We did not see a high level of expression of IL-6 and IL-1a, nor did the levels differ between the OPLL and NOPLL groups. FGF-2 was not studied. 

This could possibly be explained by the hypothesis given below. The first one is that they might be examining an earlier stage than ours. In their ligamentum flavum, the accumulation of mesenchymal cells in continuous OPLL was frequent, and there was no apparent ossification [58]. For their OPLL, the calcification front consisted of a fibrocartilage layer, a calcification front, and a calcified cartilage layer [56]. They noted that the structure was less organized in the continuous OPLL (fig. 1 in Saito et al.’s 2023 paper). In our sample, there seems to be a larger proportion of the bony components. IL-6, IL-1a, FGF-2, and RANTES are cytokines, which could be responsible for the initiation of the ossification reaction, while the factors we detected (osteogenic and anti-osteoclastogenic) were the end phenotypes. However, we did not specifically study the bony structure and associate it with disease progression, and further studies are needed to clarify this. 

When the cells were explanted from a tissue and remained in the culture for at least 48 h plus further treatment, selective growth by the use of a specific medium or the loss or gain of a phenotypic trait may happen during this period. Osteoblasts are known to secrete pro-inflammatory cytokines including IL-6, among other growth factors [59], so it cannot be ruled out that IL-6, IL-1a, and RANTES were artificially stimulated from some osteoblasts in the mixed culture. It also cannot be ruled out that some vascular stem/progenitor cells (VSCs) or fibroblasts are artificially amplified and become a source of FGF-2. Anyhow, it did demonstrate that IL-6, when added to mesenchymal cells from OPLL, causes chondrogenic and osteogenic differentiation [56]. 

Furthermore, it is possible that the Japanese populations that were studied in Yayama et al. (2022) and Saito et al. (2023) have different genetic variations from those studied in Taiwan. The Japanese population is relatively less diversified, and it may be easier for a single genetic variant to be more prominent, while those in Taiwan, having their own aboriginal people, being frequently colonized by foreign countries, and receiving populations migrating from the mainland, may be more mixed. Our data using the protein array are more of an exploratory tool and still need other complementary methods to confirm them. There remain more experiments to be conducted.

### 4.12. Concern of Limitations and Future Advances

The limitation of our study lies in the fact that the sampling of the tissue could not be unified due to the circumstances of the surgery. The size of the sample was not equal, and further trimming creates bias for different disease progressions. This may result in a difference in the protein expressed. The second limitation of the study was that the patients who were diagnosed with non-OPLL were not completely healthy and may have a certain pathology in their sample, although no patients showed evidence of congenital bone disorders, musculoligamentous tissue abnormalities, or Paget disease. This may have obscured the difference that should have been picked up or artificially created differences that were not present between OPLL and healthy patients.

We have utilized arrays that contain 80 proteins and found some interesting targets, which we will look at in detail. It is still a directed study, meaning we have in mind that we want to pick up cytokines and chemokines. The next step would be conducting an open study using differential proteomic tools to render more unexpected targets. Saito et al. (2023) gave a good example of the histological staging and morphological distinction of the ossification front. Our future study will certainly be more specific in terms of the sample size, orientation, and distinction of the histological feature. 

## 5. Conclusions

A comprehensive diagram of molecules differentially expressed between serum and the OPLL tissue was devised (Figure 7). Leptins were higher in serum from patients with OPLL. Tissues from OPLL patients were found to have significantly higher levels of AGN, OPN, and OPG. A stem cell migration-friendly environment is demonstrated by the elevated levels of MIP1 delta, MCP-1, and RANTES in the serum. A certain level of inflammation is maintained; however, it was not a full-blown inflammatory environment like those in arthritis, as demonstrated by the lack of a high level of TNF-alpha and IL-1 in the blood. Many bone-promoting cytokines were elevated (oncostatin M, FGF-9, LIF, osteopontin, osteoprotegerin, TGF-beta2), as well as a factor that inhibits osteoclastogenesis (IL-10) locally in the tissue and not in the blood. This demonstrated a state of robust bone growth without the balance of osteoclastogenesis. Molecules promoting angiogenesis, including angiotensin, vEGF, and osteoprotegerin, are abundant in the OPLL tissue, which paves the way for robust bone growth. Serum leptins in OPLL patients are significantly higher than those in NOPLL patients, corroborating others’ observations that they may serve as disease markers. However, the local leptin within the tissue may not be directly responsible for the disease mechanism, as the level was low and not regulated between OPLL and non-OPLL. Another component of the insulin/IGF/adipocyte axis, IGFBPs, is highly expressed locally in OPLL and may contribute to the bone growth by regulating the available IGF-1 to the osteocytes.

## Figures and Tables

**Figure 1 biomedicines-11-02021-f001:**
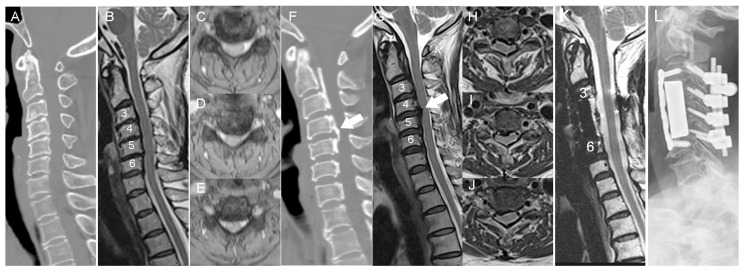
(**A**) Sagittal view of a CAT scan of a 60-year-old female with NOPLL and spinal stenosis at C3-4-5-6. (**B**) Sagittal view of T2-weighted images from an MRI of the same patient. (**C**–**E**) Axial views of T2-weighted images from an MRI at the levels of C3-4-5-6. Spinal stenosis with compressive myelopathy was noted. (**F**) Sagittal view of a CAT scan of a 67-year-old female with OPLL and severe spinal stenosis at C3-4-5-6. (**G**) Sagittal view of T2-weighted images from an MRI of the same patient. The worst compression by OPLL was at the C4-5 level. (**H**–**J**) Axial views of T2-weighted images from an MRI at the levels of C3-4-5-6. Spinal stenosis with compressive myelopathy was noted. (**K**) Sagittal view of T2-weighted images from an MRI of the same patient after surgery. OPLL was resected, and a large CSF space was noted, compared with the pre-op MRI. (**L**) Radiograph of a lateral view, post-op. Corpectomy at C4-6 and laminectomy at C3-7 from the anterior, posterior procedures of laminectomy at C1-7, and fixation at C3-6. Arrow, OPLL; Star, CSF space.

**Figure 2 biomedicines-11-02021-f002:**
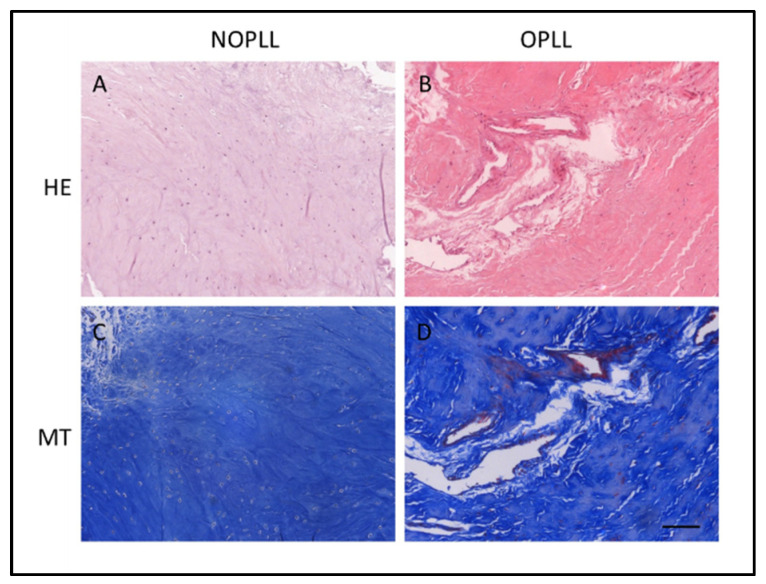
Hematoxylin-Eosin (HE) staining and Masson’s Trichrome (MT) staining of tissue of NOPLL (**A**,**C**) and OPLL patients (**B**,**D**). Scale bar: 50 μm.

**Figure 3 biomedicines-11-02021-f003:**
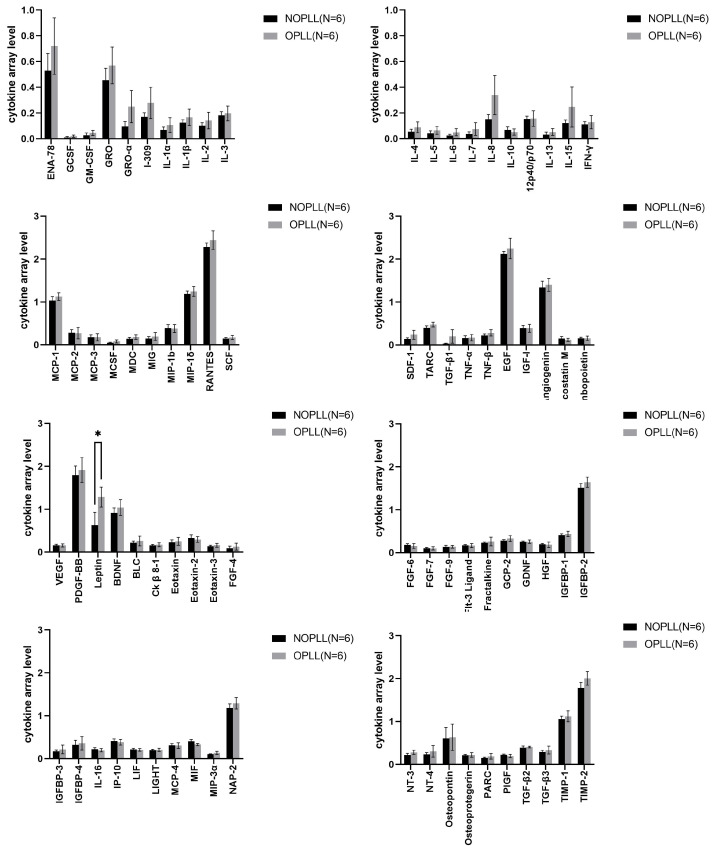
Comparison of serum samples from patients with NOPLL and OPLL by the Human Cytokine Array. The results of the Human Cytokine Array demonstrated the significantly higher intensity of leptin in OPLL samples compared to that in NOPLL samples. The leptin was significantly higher in the OPLL serum sample by Šídák’s multiple comparisons test. Data are presented as the mean ± SEM, *: *p* < 0 005.

**Figure 4 biomedicines-11-02021-f004:**
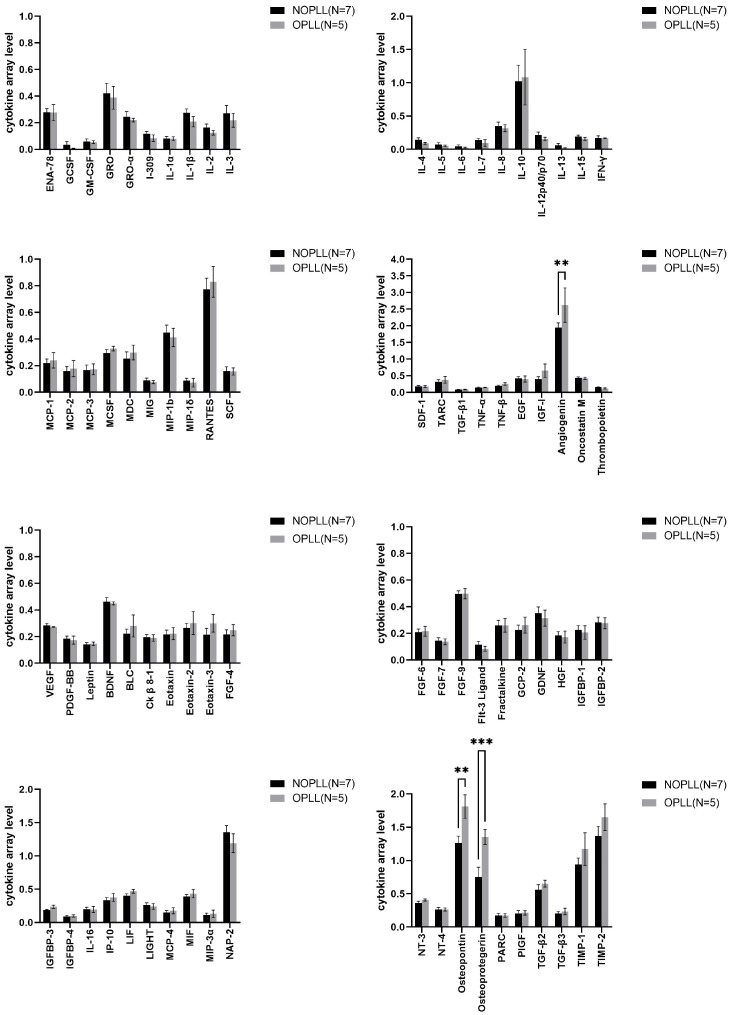
Comparison of tissue samples from patients with NOPLL and OPLL by the Human Cytokine Array. The results of the Human Cytokine Array demonstrated the significantly higher intensity of ANG, OPN, and OPG in OPLL samples compared to that in NOPLL samples. These proteins were significantly higher in OPLL tissue by Šídák’s multiple comparisons test. Data are presented as the mean ± SEM, **: *p* < 0.01; ***: *p* < 0.001.

**Figure 5 biomedicines-11-02021-f005:**
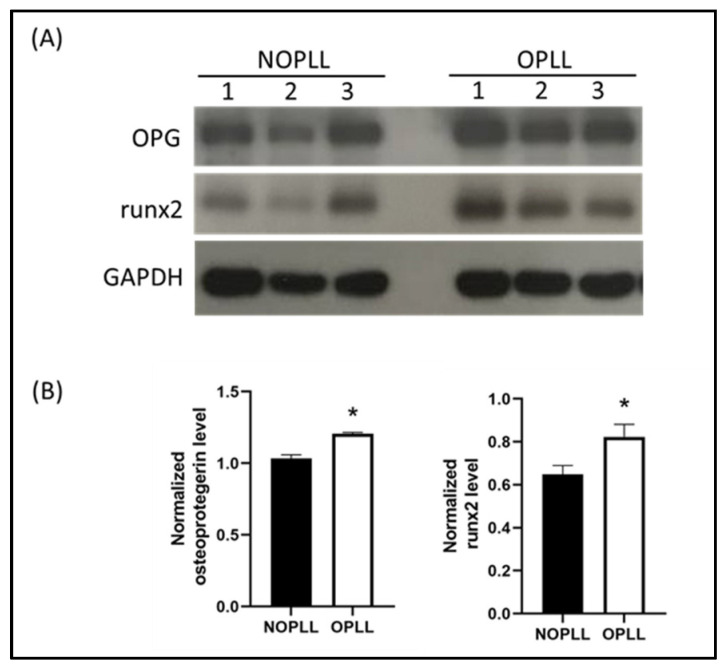
(**A**) A representative case of immunoblot analysis. (**B**) Immunoblot analysis demonstrating the relative expression of OPG and Runx2 compared with GAPDH (*: *p* < 0.05, compared to NOPLL).

**Figure 6 biomedicines-11-02021-f006:**
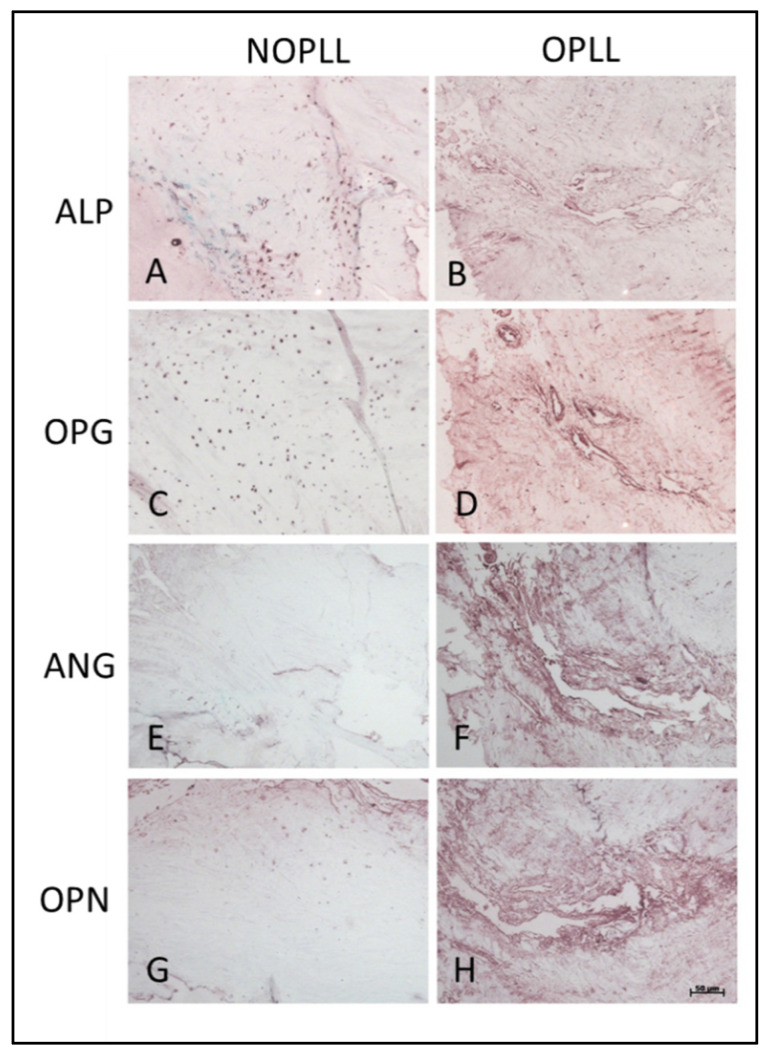
Immunohistochemical staining of (**A**,**B**) Alkaline phosphatase (ALP), (**C**,**D**), Osteoprotegerin (OPG), and (**E**,**F**) Angiogenin (ANG), and (**G**,**H**) Osteopontin (OPN) staining of NOPLL and OPLL tissue. Scale bar, 50 μm.

**Figure 7 biomedicines-11-02021-f007:**
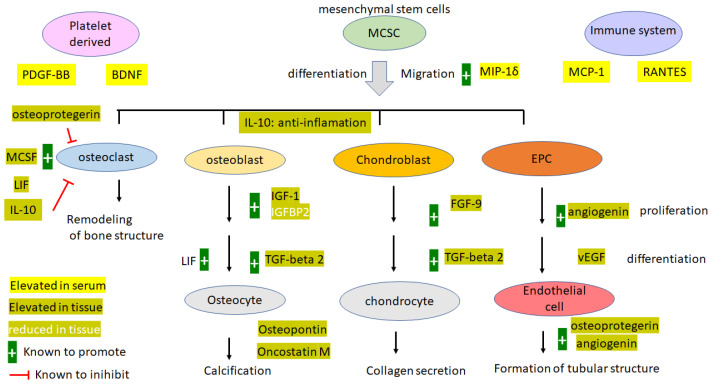
Graphical summary of the protein expressed.

**Table 1 biomedicines-11-02021-t001:** Comparison of NOPLL and OPLL patients.

	NOPLL	OPLL	*p* Value
patient numbers	33	18	
age	54.7 ± 10.2	59.7 ± 9.7	0.09
male (%)	15 (45.5)	13 (72.2)	0.08
lesion level ^#^			
one	12	6	
two	10	11	
three	10	1	
four	1	0	
K-line			
positive	n/a	13	
negative	n/a	5	
OPLL type			
segmental	n/a	6	
continuous	n/a	2	
mixed	n/a	10	
JOA score ^##^			
pre-op	13.8 ± 2.5	12.9 ± 3.3	0.42
1 mo	14.4 ± 2.5 **	11.3 ± 4.7	0.06
3 mo	14.5 ± 2.3 **	11.8 ± 3.4	0.05
6 mo	14.5 ± 3.1 **	13.5 ± 2.7	0.07

Values are the mean ± standard deviation; OPLL, Ossification of Posterior Longitudinal Ligament; NOPLL, Non-OPLL; **, *p* < 0.05, statistically significant improvement, compared to pre-op status; n/a, not available; ^#^ lesion level, discs in NOPLL, bodies in OPLL; ^##^ JOA score, Japanese Orthopedic Association score.

**Table 2 biomedicines-11-02021-t002:** Comparison of protein levels in NOPLL and OPLL patients.

			*p* Value
Sample Type	NOPLL	OPLL	
Serum leptin	0.40 ± 0.25	1.23 ± 0.23	0.0138 *
Tissue leptin	0.141 ± 0.037	0.146± 0.028	0.8248
Tissue angiogenin	1.94 ± 0.15	3.0 ± 0.50	0.0012 **
Tissue osteopontin	1.27 ± 0.10	1.81 ± 0.18	0.0038 **
Tissue osteoprotegerin	0.75 ± 0.15	1.35 ± 0.11	0.0006 **

Values are the mean ± standard deviation; OPLL, Ossification of Posterior Longitudinal Ligament; NOPLL, Non-OPLL; *, *p* < 0.05, intergroup statistically significant; **: *p* < 0.01.

**Table 3 biomedicines-11-02021-t003:** Proteins that have higher expression in serum than in tissue.

Protein Name	Level (Serum)	Level (Tissue)	Ratio (Serum/Tissue)	*p* Value	Comments
MCP-1	1.13	0.24	4.69	<0.0001	injury elicite, recruite lymphocyte
MIP-1δ	1.25	0.07	17.3	<0.0001	enhances osteoclastogenesis
EGF	2.25	0.40	5.62	<0.0001	platelet-derived factors
RANTES	2.44	0.83	2.94	0.0002	T cell-regulated, late expression, for maitaining inflammation
PDGF-BB	1.92	0.17	11.0	0.0004	platelete-derived factors
Leptin	1.28	0.15	8.79	0.0012	serum leptin high = OPLL marker
BDNF	1.04	0.45	2.32	0.0168	platelet-derived factors
IGFBP-1	0.44	0.21	2.12	0.0206	regulate IGF-1 bioavaialbility
IGFBP-2	1.64	0.28	5.95	<0.0001	regulate IGF-1 bioavaialbility

**Table 4 biomedicines-11-02021-t004:** Proteins that have higher expression in tissue than in serum.

Protein Name	Level (Serum)	Level (Tissue)	Ratio (Tissue/Serum)	*p* Value	Comments
IL-10	0.05	1.08	20.98	0.02	cytokine for general repair, inhibit osteoclastogenesis
MCSF	0.08	0.328	4.10	0.0002	promote osteoclastgenesis
Angiogenin	1.40	2.62	1.87	0.0353	promoting angiogenesis
Oncostatin M	0.12	0.42	3.50	0.0002	pleiotropic regulator of bone formation and resorption
VEGF	0.16	0.27	1.74	0.0153	essential growth factor for vascular endothelial cells
FGF-9	0.14	0.50	3.60	<0.0001	promote chondrogenesis and osteogenesis
LIF	0.21	0.47	2.29	0.0002	action on skeletal cells, osteoclasts secrete LIF to promote abnormal bone remodeling
Osteopontin	0.63	1.81	2.86	0.0122	organic component of bone, mineralization, bone remodeling
Osteoprotegerin	0.22	1.35	6.15	<0.0001	a decoy receptor for RANKL, suppress osteoclastogenesis and bone resorption
TGF-β2	0.41	0.65	1.61	0.0009	important in bone homeostasis

## Data Availability

Data available on request due to privacy restrictions and subject to IRB approval.

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
