# Peer review of "Comparative Study of the Cytokine Profiles of Serum and Tissues from Patients with the Ossification of the Posterior Longitudinal Ligament"

_biomedicines, 2023, doi:10.3390/biomedicines11072021_

Round 1

Reviewer 1 Report

Authors present a study on 51 patients with (18) and without ossification of the posterior longitudal ligament (OPLL) to compare levels of proinflammatory cytokines in the ligament and serum levels between the two groups. In the tissue angiogenin (ANG), osteopontin (OPN), and osteopro-tegerin (OPG) were higher in the OPLL group with upregulation of chemotactic cytokines in the serum and osteoclastogenesis inhibitors in the ligament tissue.  Molecules promoting angiogenesis including  are abundant in the OPLL tissue which enables bone growth.

Low number of patients and retrospective character are main drawbacks of the study.

Please report your findings on IL-6 levels, which seems to be a hot topic at the moment. IL-1α, basic fibroblast growth factor, and RANTES are further interesting cytokines which are found to have a signficance for OPLL and hyperostosis in general.

Two very important studies are missing from the literature review, I suggest to include and compare the results and comment extensively, especially on the role of IL-6:

Saito H, Yayama T, Mori K, Kumagai K, Fujikawa H, Chosei Y, Imai S. Increased Cellular Expression of Interleukin-6 in Patients With Ossification of the Posterior Longitudinal Ligament. Spine (Phila Pa 1976). 2023 Mar 15;48(6):E78-E86. doi: 10.1097/BRS.0000000000004557. Epub 2022 Dec 5. PMID: 36729990.

Yayama T, Mori K, Saito H, Fujikawa H, Kitagawa M, Okumura N, Nishizawa K, Nakamura A, Kumagai K, Mimura T, Imai S. Cytokine Profile From the Ligamentum Flavum in Patients with Ossification of the Posterior Longitudinal Ligament in the Cervical Spine. Spine (Phila Pa 1976). 2022 Feb 1;47(3):277-285. doi: 10.1097/BRS.0000000000004302. PMID: 34919077.

I suggest to include future advances section and possible clinical application of findings of this study.

Did you find any connection to TGF-beta-1?:

Han IB, Ropper AE, Jeon YJ, Park HS, Shin DA, Teng YD, Kuh SU, Kim NK. Association of transforming growth factor-beta 1 gene polymorphism with genetic susceptibility to ossification of the posterior longitudinal ligament in Korean patients. Genet Mol Res. 2013 Feb 28;12(4):4807-16. doi: 10.4238/2013.February.28.26. PMID: 23479171.

Were there any differences which were age- or gender related? Were there any patients who had a known metabolic disease or the disease of the metabolism of the bone or calcium which could have influenced the results?

Acceptable.

Reviewer 2 Report

General impression

In this research, the authors investigated the cytokine profiles of serum and tissues from patients with ossification of posterior longitudinal ligament (OPLL) and compared them with non OPLL patients.  And they concluded that some cytokines are abundant in the OPLL tissue, which pave the way for bone growth.  As the authors mentioned, I think there was no study about tissue biomarkers.  So, this report must be valuable elucidating for causes of OPLL.

The methodology of this study was precisely explained.  Also, the limitations of this project were indicated.  I could not find either error in writings or mistakes in the text.  However, I have some minor requests to be revised as stated below.  After they have been resolved, I will judge this manuscript can be accepted and published by biomedicines journal.

*I ask the authors that correction parts will be shown in red color in the revised manuscript.

1. Figure 1  page 3 line 109-110

Details of surgical procedures should be more clearly indicated.

  I guess this case was performed anterior procedures of corpectomy at C4-6 and fixation at C3-7, and posterior procedures of laminectomy at C1-7 and fixation at C3-6.  They are different from the explanation of legend.

2. Figure 7  page 15 line 504

  I cannot find the Figure 7.

3. limitation

I guess this study may have some limitations.  I request the authors will mention them independently in the end of discussion section.

Round 2

Reviewer 1 Report

Authors have sufficiently responded to reviewer remarks, include the authors response into the text of the manuscript fully. 

Ok.